# Modeling the Solvent Extraction of Cadmium(II) from Aqueous Chloride Solutions by 2-pyridyl Ketoximes: A Coordination Chemistry Approach

**DOI:** 10.3390/molecules24122219

**Published:** 2019-06-13

**Authors:** Eleni C. Mazarakioti, Amaia Soto Beobide, Varvara Angelidou, Constantinos G. Efthymiou, Aris Terzis, Vassilis Psycharis, George A. Voyiatzis, Spyros P. Perlepes

**Affiliations:** 1Department of Chemistry, University of Patras, 265 04 Patras, Greece; eleni.mazarakioti@uv.es (E.C.M.); varvara_angelidou@hotmail.com (V.A.); dinosef@yahoo.com (C.G.E.); 2Foundation for Research and Technology-Hellas (FORTH), Institute of Chemical Engineering Sciences (ICE-HT), Platani, P.O. Box 1414, 265 04 Patras, Greece; asoto@iceht.forth.gr; 3Institute of Nanoscience and Nanotechnology, NCSR “Demokritos”, 153 10 Aghia Paraskevi Attikis, Greece; a.terzis@inn.demokritos.gr

**Keywords:** cadmium(II), coordination chemistry, model studies for liquid-liquid extraction, phenyl 2-pyridyl ketoxime as ligand, 2-pyridyl ketoximes as cadmium(II) extractants, Raman spectra, single-crystal X-ray structures

## Abstract

The goal of this work is to model the nature of the chemical species [CdCl_2_(extractant)_2_] that are formed during the solvent (or liquid-liquid) extraction of the toxic cadmium(II) from chloride-containing aqueous media using hydrophobic 2-pyridyl ketoximes as extractants. Our coordination chemistry approach involves the study of the reactions between cadmium(II) chloride dihydrate and phenyl 2-pyridyl ketoxime (phpaoH) in water-containing acetone. The reactions have provided access to complexes [CdCl_2_(phpaoH)_2_]∙H_2_O (**1**∙H_2_O) and {[CdCl_2_(phpaoH)]}_n_ (**2**); the solid-state structures of which have been determined by single-crystal X-ray crystallography. In both complexes, phpaoH behaves as an *N*,*N*’-bidentate chelating ligand. The complexes have been characterized by solid-state IR and Raman spectra, and by solution ^1^H NMR spectra. The preparation and characterization of **1**∙H_2_O provide strong evidence for the existence of the species [CdCl_2_(extractant)_2_] that have been proposed to be formed during the liquid-liquid extraction process of Cd(II), allowing the efficient transfer of the toxic metal ion from the aqueous phase into the organic phase.

## 1. Introduction

The rapid industrialization of our society during the 20th century provided humans with a better quality of life, but it has simultaneously generated a series of environmental issues; one serious problem is associated with the accumulation of heavy toxic metal ions in industrial effluents [1]. Cadmium(II) is a very toxic metal ion, being introduced into the environment by anthropogenic activities, such as the production of alkaline batteries, lead-zinc mining, photography, electroplate, and pigments [2], thus affecting plants and humans. Exposure of humans to elevated Cd(II) concentrations causes several acute and chronic harmful symptoms in the liver, kidneys, and cardiovascular and nervous systems [1]. Diseases caused by Cd(II) toxicity are proteinuria, aminoaciduria, cadmium-promoted creatinuria, cadmium-induced renal tubular dysfunction, and a form of itai-itai, the latter arising from drinking water contaminated with this metal ion and leading to osteomalacia and bone decalcification [3]. These macroscopic symptoms require the presence of soluble forms of Cd(II), which are mobilizable into waters, the food chain, and into the cellular fluids of organisms [4]. The toxicity of this metal ion is mainly attributed to its strong interactions with the cysteinyl sulfur atoms of enzymes and proteins [5]. Due to the non-biodegradable and non-decomposable nature of Cd(II), efficient methods for its removal from wastewater should be developed. Several methods for this use have been designed and developed, such as ion exchange, chemical precipitation, membrane separation, adsorption methods, and solvent extraction [1,6,7]. This work is related to the latter method.

Solvent extraction is increasingly used for both the recovery of useful metals and the removal of toxic metal ions. In a typical flow sheet of the former process [8], the metal contents in an ore are leached into an aqueous solution, and the valuable metal ion is then transferred to a water-immiscible solvent where the unit operations of concentration and separation take place. Stripping back to provide a pure electrolyte allows the recovery of the metal by electroreduction. The removal of toxic metal ions using solvent extraction occurs through the complexation of the metal ion with an organic ligand to form a species that is transferred from the aqueous to the organic phase in a two-phase system [9]. Hexane and kerosene are mainly used in industry due to their low toxicity, but most laboratory studies have been carried out with chloroform as the organic phase. Carbon dioxide is also used for the solid-liquid extraction of heavy metal ions from contaminated soils. There are three types of *liquid-liquid* extraction: (a) The case where the extractant, i.e., the organic ligand, and the metal ion source are both soluble in the aqueous phase, and the extracted complex is soluble in the organic phase; (b) the case where both the extractant and the metal ion source are insoluble in the aqueous phase and the complexation takes place at the interphase surface, and the metal species is then transferred into the organic phase; (c) the case where the extractant is soluble only in the organic phase and the metal ion source is soluble only in the aqueous phase, the complexation reaction occurring again at the interphase surface before the transfer of the metal ion-extractant complex into the organic phase.

Most efficient extractants are chelating or macrocyclic ligands; the reason for this is the high thermodynamic stability of the resulting metal complexes due to the chelate or macrocyclic effects [10]. After decades of research, it is now well established that the main requirements for an effective extractant are the following: (i) Fast binding of the toxic metal ion by the extractant; (ii) good stability of the metal ion-extractant complex against hydrolysis; (iii) selective metal ion complexation with no or weak affinity for group 1 (alkali ions) and 2 (alkaline earth ions) metal ions, which are present at high concentrations in natural and waste waters; (iv) high enough binding strength for the metal ion to be extracted; (v) reversible complexation reaction, which allows for the complete recovery of the metal ion without significant extractant decomposition or destruction.

Several extractants have been used for the liquid-liquid extraction of toxic Cd(II), including [9,11,12] organophosphorus derivatives, dithiocarbamates, xanthates, EDTA derivatives, crown ethers, azacrowns, derivatized calix[4]arenes, substituted 8-quinolols, pyridine carboxamides, and *pyridyl ketoximes*. In an excellent study, which has been the stimulus of the present work, Parus and co-workers investigated the solvent extraction of Cd(II) from aqueous chloride solutions using 1-(2-pyridyl)-trideca-1-one oxime (2PC12) and 1-(2-pyridyl)-pentadecane-1-one oxime (2PC14) [12]. The structural formulae of the extractants are presented in Figure 1. The influence of the extractant, metal ion and chloride ion concentration, and the nature of various polar and non-polar solvents (diluents) on the extraction efficiency was studied in detail. Cadmium(II) was effectively extracted using chloroform (CHCl_3_) or hydrocarbons mixed with decan-1-ol as organic phases, and stripped from the loaded organic phase with water and aqueous ammonia solutions. Plotting (including log-log plots) the effect of the varying concentrations of 2PC12 and 2PC14 on the extraction capability, the authors concluded that the neutral species [CdCl_2_(extractant)_2_] were formed during the liquid-liquid extraction process, allowing the transfer of the complexed toxic metal ion into the organic phase. 2-pyridyl ketoximes (including 2PC12) have also been used for copper(II) extraction from chloride solutions [13].

We have embarked on a new program [14] aiming at modeling various aspects of the liquid-liquid extraction of toxic Cd(II) from chloride solutions by 2PC12 and 2PC14 [12], adopting an inorganic (coordination) chemistry approach. In this work, which is the first of a series of papers, we were interested in investigating the existence of the [CdCl_2_(2PC12)_2_] and [CdCl_2_(2PC14)_2_] species that have been proposed to form during the solvent extraction process. The reactions of 2PC12 or 2PC14 with CdCl_2_∙2H_2_O in various organic solvents (ethanol, EtOH; acetonitrile, MeCN; CHCl_3_) or organic solvent mixtures (EtOH/MeCN, MeCN/CHCl_3_) gave solid products that could not be crystallized for single-crystal X-ray studies. Thus, we used phenyl 2-pyridyl ketoxime [other names: phenyl(pyridin-2-yl)methanone oxime or (N-hydroxy-1-phenyl-2-yl)methanimine] in our synthetic efforts, which gave crystalline complexes. This compound (Figure 1, its abbreviation will be phpaoH in the present work) is a satisfactory analog (albeit not an ideal one) of the extractants 2PC12 and 2PC14. The three ketoximes possess a 2-pyridyl ring, an oxime group, and a hydrophobic substituent on the oxime carbon; the main difference is the presence of a long aliphatic chain in the real extractants instead of the phenyl aromatic ring substituent in phpaoH. This paper describes our results from the synthetic investigation of the general reaction system CdCl_2_∙2H_2_O/phpaoH and from the full structural and spectroscopic characterization of the products; the implication of our study with respect to the solvent extraction of Cd(II) from aqueous chloride solutions by 2-pyridyl ketoximes have also been critically discussed. The present work can be considered as a continuation of our interest in several aspects of Cd(II) chemistry [15,16,17,18,19,20] and the coordination chemistry of 2-pyridyl oximes (aldo-, keto-, and amidoximes) [20,21,22,23,24,25,26,27,28,29,30,31,32,33]. Our previous experience on the latter area shows that phpaoH behaves similarly with methyl 2-pyridyl ketoxime (mepaoH; the substituent on the oxime carbon atom is a methyl group) [21,30] in 3d- and mixed 3d/4f-metal complexes, and this justifies partly the choice of phpaoH for our model studies.

## 2. Results and Discussion

### 2.1. Synthetic Comments

A variety of Cd(II)/Cl^−^/phpaoH reaction systems involving various metal sources, and different reagent ratios, solvent media, and crystallization methods were systematically employed before arriving at the optimized synthetic conditions reported in Section 3. Since all the extraction experiments were carried out in an H_2_O-organic solvent system [12], we used solvent mixtures comprising both H_2_O and an organic solvent, the latter being mixed well with the former. It was not possible to use a two-phase system, e.g., H_2_O-CHCl_3_ (as in the real extraction experiments [12]), because of the rather poor solubility of phpaoH in the organic phase. In all the extraction experiments, the pH of the aqueous phases was between 3.5 and 3.8 [12], suggesting that 2PC12 and 2PC14 remain neutral during the process; we thus avoided the addition of an external base (e.g., LiOH, Et_3_N, R_4_NOH, etc.) in the reaction systems. The only solvent mixture that gave crystals of the products (suitable for single-crystal X-ray crystallography) was the H_2_O-Me_2_CO one. Depending on the reactants molar ratio, two different CdCl_2_/phpaoH products were obtained.

The 1:2 reaction between CdCl_2_∙2H_2_O and phpaoH in H_2_O-Me_2_CO (1:1 v/v) gave a colorless solution from which crystals of [CdCl_2_(phpaoH)_2_]∙H_2_O (**1**∙H_2_O) were subsequently isolated in rather low yields (~30%). Our efforts to increase the yield by increasing the phpaoH:Cd(II) molar ratio from 2:1 to 3:1 or/and by increasing the H_2_O volume percentage in the solvent mixture resulted in **1**∙H_2_O contaminated with free phpaoH (analytical and IR evidence). Increase of the Me_2_CO volume percentage did not significantly improve the yield. Assuming that the mononuclear complex is the only product from the reaction system, its formation is summarized by chemical Equation (1).
(1)$${\mathrm{CdCl}}_{2}\cdot 2H_{2}O\;+2\;\mathrm{phpaoH}\overset{H_{2}O-Me_{2}CO}{\to }\;[{\mathrm{CdCl}}_{2}(\mathrm{phpaoH}{)}_{2}]\;+\;2\;H_{2}O$$
As mentioned above, the CdCl_2_∙2H_2_O:phpaoH molar ratio affected the product identity. The 1:1 reaction between CdCl_2_∙2H_2_O and phpaoH in the same solvent mixture used for the preparation of **1**∙H_2_O, i.e., H_2_O-Me_2_CO (1:1 v/v), gave a microcrystalline powder **A** whose analytical data and IR spectra were different from those of **1**∙H_2_O; the analytical data fitted well the empirical formula CdCl_2_(phpaoH), suggesting the existence of a 1:1 complex. Keeping constant the solvent mixture, all our efforts to obtain crystals of this 1:1 product were in vain. After hundreds of experiments, the solution of the problem came from a non-orthodox reaction, which involved the use of a chloride-free Cd(II) source and “external” chloride ions. The 1:1:2 Cd(ClO_4_)_2_∙6H_2_O/phpaoH/H_3_NOH^+^Cl^−^ (hydroxylamine hydrochloride) reaction mixture in H_2_O-Me_2_CO (1:5 *v*/*v*) gave a pale yellow solution, which—upon standing undisturbed—gave well-formed colorless crystals of the polymeric complex {[CdCl_2_(phpaoH)]}_n_ (**2**) in high yield (~80%). Use of a solvent mixture with a higher H_2_O volume percentage, e.g., H_2_O-Me_2_CO = 1:1, decreased the yield of the reaction and made the quality of single crystals worse. The IR spectrum of **2** was identical with that of **A,** proving that the polymeric compound can also be prepared (albeit in the form of a microcrystalline powder) by using CdCl_2_∙2H_2_O as the source of Cd(II). With the results at hand, it is rather difficult to estimate the role of ClO_4_^−^ ions and hydroxylamine hydrochloride in the formation of product with good crystallinity; we tentatively propose that the higher ionic strength of the Cd(ClO_4_)_2_∙6H_2_O/phpaoH/H_3_NOH^+^Cl^−^ reaction medium has a positive impact on the quality of the crystals obtained. The formation of the complex, using the method that gave single crystals, is illustrated in the chemical Equation (2).
(2)$$n\;\mathrm{Cd}({\mathrm{ClO}}_{4}{)}_{2}\cdot 6H_{2}O+n\;\mathrm{phpaoH}+2nH_{3}{\mathrm{NOH}}^{+}{\mathrm{Cl}}^{-}\overset{H_{2}O-Me_{2}CO}{\to }\{[{\mathrm{CdCl}}_{2}(\mathrm{phpaoH})]{\}}_{n}+2n\;\;H_{3}{\mathrm{NOH}}^{+}({\mathrm{ClO}}_{4}){}^{-}+6nH_{2}O$$

### 2.2. Description of Structures

The structures of **1**∙H_2_O and **2** were determined by single-crystal X-ray crystallography. Crystallographic data are listed in Table 1. Various structural plots are shown in Figure 2, Figure 3, Figure 4, Figure 5 and Figure 6. Selected interatomic distances and angles are given in Table 2 and Table 3, while hydrogen bonding details are summarized in Table 4 and Table 5.

The crystal structure of **1∙**H_2_O consists of complex molecules [CdCl_2_(phpaoH)_2_] and lattice H_2_O molecules in a 1:1 ratio. The Cd^II^ atom is coordinated by two chloro (or chlorido) groups (Cl1, Cl2), two oxime nitrogen atoms (N2, N12), and two 2-pyridyl nitrogen atoms (N1, N11), the latter four arising from two *N*,*N*’-bidentate chelating (or 1.011 adopting the Harris notation [34]) phpaoH ligands. The coordination geometry of the metal ion is distorted octahedral, the *trans* and *cis* donor atom -Cd^II^- donor atom bond angles being in the ranges 149.4(1)–166.2(1)° and 67.3(1)–105.3(1)°, respectively. The distortion from the regular octahedral geometry is primarily attributed to the small bite angles of the two 5-membered chelating rings [N1-Cd1-N2 = 69.8(1)°, N11-Cd1-N12 = 67.3(1)°]. The octahedral molecule is the *cis-cis-cis* isomer considering the positions of the chloro ligands, the oxime, and the 2-pyridyl nitrogen donor atoms. There is one intramolecular hydrogen bond of medium strength with the uncoordinated oxime oxygen atom O11 as a donor and the coordinated chloride Cl1 as acceptor. Through the O1W-H_A_(O1W)∙∙∙Cl2 “intramolecular” hydrogen bond, each lattice H_2_O molecule is associated with one [CdCl_2_(phpaoH)_2_] molecule and molecular {[CdCl_2_(phpaoH)_2_]-H_2_O} pairs are thus formed in the crystal; this view is emphasized in Figure 2.

The complex [CdCl_2_(phpaoH)_2_] and solvent H_2_O molecules form double chains (1D), which are parallel to the *b* axis (Figure 3) through O1-H(O1)∙∙∙O1W, O1W-H_A_(O1W)∙∙∙Cl2, and O1W-H_B_(O1W)∙∙∙Cl1 H bonds. Each lattice H_2_O is connected to three complex molecules.

The double chains in the structure of **1**∙H_2_O are further linked through π-π stacking interactions between centrosymmetrically-related 2-pyridyl rings containing the N1 atom, forming layers parallel to the (001) plane (Figure 4). The centroid∙∙∙centroid distance between the parallel aromatic rings is 3.761 Å.

Compound **2** is a 1D coordination polymer. Its crystal structure consists of zigzag chains extended parallel to the crystallographic ‘a’ axis (Figure 5). The Cd^II^ atoms are doubly bridged by two asymmetric μ-chloro groups. One *N*,*N*’-bidentate chelating (1.011) phpaoH ligand completes six-coordination at each metal ion. The Cd^II^ coordination polyhedron is a distorted octahedron. The *trans* coordination angles are in the 152.0 (1)–168.4 (1)° range, while the *cis* ones are in the 67.6 (1)–112.6 (1)° range. There are rather weak, intrachain hydrogen bonds in which the oxime oxygen atom O1 is the donor and the bridging chloro groups Cl2 and Cl1 are the acceptors; only the O1-H(O1)∙∙∙Cl2 component of this bifurcated hydrogen bond is shown in Figure 5.

The 1D polymeric chains in the crystal structure of **2** are further linked through weak π-π stacking interactions between centrosymmetrically-related 2-pyridyl rings, forming layers that extend parallel to the (001) plane (Figure 6). The centroid∙∙∙centroid distance between the aromatic heterocyclic rings is 4.079 Å. In addition, there is a C-H∙∙∙π interaction, in which a pyridyl carbon atom (C3) is the donor and the phenyl ring (C7–C12) of phpaoH is the acceptor.

The Cd-Cl bond lengths in **1∙**H_2_O [2.532(1) and 2.534(1) Å] are shorter than those in **2** [2.564(1)–2.686(1) Å] due to the terminal character of the chloro ligands in the former versus the bridging one in the latter. The bridging Cd-Cl bond distances in **2** are typical for other six-coordinate Cd(II) complexes containing bridging chloro groups [15]. The carbon-nitrogen [1.278(3)–1.293(5) Å] and nitrogen-oxygen [1.381(3)–1.387(3) Å] bond lengths of the coordinated oxime groups are similar (for a given bond type) in the two complexes; the nitrogen-oxygen bonds are longer than the carbon-nitrogen bonds due to their different multiplicity (single versus double). The carbon-nitrogen and nitrogen-oxygen bond lengths in the two complexes are practically similar and slightly larger, respectively, compared with the corresponding bond distances observed in the crystal structures of the various polymorphs of the free phpaoH [35,36,37].

Complexes **1**∙H_2_O and **2** are new members in the large family of metal complexes with the neutral phpaoH or its deprotonated form (phpao^−^) as ligands [21,24,27,29,30,32,38,39,40,41,42,43,44,45,46]; only representative references are listed. The only structurally characterized Cd(II)/phpaoH species to-date is the cationic octahedral complex [Cd(phpaoH)_3_](NO_3_)_2_ [46], prepared by the 1:2 reaction of Cd(NO_3_)_2_∙4H_2_O and phpaoH. The Cd-N bond lengths [2.320(3)–2.402(3) Å] in this complex are similar with those of **1**∙H_2_O and **2** [2.323(2)–2.509(2) Å]. The polymeric structural type of **2** is completely new in the coordination chemistry of phpaoH. There have been only two mononuclear complexes reported with the formula [M^II^Cl_2_(phpaoH)_2_] (M^II^ = a divalent metal), namely [NiCl_2_(phpaoH)_2_]∙Me_2_CO [42] and [MnCl_2_(phpaoH)_2_]∙0.02EtOH [43]; both compounds contain *cis-trans-cis* octahedral molecules considering the positions of the chloro groups, the oxime nitrogen atoms, and the 2-pyridyl nitrogen atoms, respectively. Complex **1,** with a *cis-cis-cis* disposition of the chemically similar donors, is thus a unique geometrical isomer in the [M^II^Cl_2_(phpaoH)_2_] series.

### 2.3. Spectroscopic Studies

We discuss first the vibrational spectra of **1**∙H_2_O and **2**. The FT-Raman spectra of the free compound phpaoH and its Cd(II) complexes **1**∙H_2_O and **2** are presented in Figure 7.

In the solid-state (KBr) IR spectrum of **1**∙H_2_O, the medium-intensity band at 3506 cm^−1^ and the weak band at 1628 cm^−1^ are attributed to the *ν*(OH) and *δ*(HOH) vibrations, respectively, of the lattice water that is present in the complex [47]. The rather high wavenumber of the stretching vibration is indicative of the non-coordinating nature of the H_2_O molecule. These bands are absent from the IR spectra of phpaoH and **2**. The presence of neutral oxime groups in the complexes is manifested by the appearance of a medium-to-strong IR band at 3384 (**1**∙H_2_O) and 3388 (**2**) cm^−1^ assigned to the *ν*(OH) vibration [48,49]. The corresponding band in the spectrum of the free ligand appears at 3154 cm^−1^. The large wavenumber difference can be explained by the involvement of the −OH group in hydrogen bonds of different strength in the complexes and in the free phpaoH compound. As expected, the *ν*(OH) peaks are hardly seen in the Raman spectra. On the contrary, at the high-frequency part of the Raman spectra, the strong peaks at 3060 (**1**∙H_2_O) and 3064 (**2**) are assigned to a *ν*(C-H) vibration [50]. The medium-intensity bands at 1568 and 1094 cm^−1^ in the IR spectrum of the free ligand phpaoH have been assigned [40,48] to the *ν*(C=N) and *ν*(N-O) vibrations of the oxime group, respectively. The 1094 cm^−1^ band is shifted to a lower wavenumber in the spectra of the complexes (1054 cm^−1^ in **1**∙H_2_O, 1042 cm^−1^ in **2**). This shift has been attributed to the coordination of the neutral oxime nitrogen [40,48]. To our surprise, the 1568 cm^−1^ band is shifted to a higher wavenumber in the IR spectra of the complexes (1590 cm^−1^ in **1**∙H_2_O, 1600 cm^−1^ in **2**), overlapping with an aromatic stretching vibration [48]. This experimental fact is not unusual [48]. Extensive studies on complexes with ligands containing a C=N bond (with the carbon atom attached to an aromatic ring) have shown [51] that a change in the s character of the N lone pair occurs upon coordination and the s character of the nitrogen orbital involved in the C=N bond increases; this change in hybridization leads to a greater C=N stretching force constant relative to the free neutral ligands, thus shifting the *ν*(C=N) band in the spectra of the complexes to higher wavenumbers. The Raman *ν*(C=N) peaks for phpaoH, **1**∙H_2_O, and **2** appear at 1599, 1626, and 1631 cm^−1^, respectively [50,52,53], while the *ν*(N-O) peaks appear at 1097 (phpaoH), 1053 (**1**∙H_2_O), and 1052 (**2**) cm^−1^, respectively [53]. The Raman coordination shifts are in the same directions with the corresponding IR ones. The medium-intensity peaks in the Raman spectra of the free ligand and the two complexes at ~1330 cm^−1^ are attributed to the NOH in-plane deformation, *δ*(NOH) [50].

The in-plane deformation band of the 2-pyridyl ring, *δ*(py), in the IR spectrum of free phpaoH at 622 cm^−1^ shifts to higher wavenumbers (~660 cm^−1^) in the spectra of the complexes, confirming the participation of the heterocyclic nitrogen atom in coordination [48]. This mode appears [50] at 622, 636, and 639 cm^−1^ in the Raman spectra of phpaoH, **1**∙H_2_O, and **2**, respectively. The Raman peaks in the 410–210 cm^−1^ region are associated with the Cd-Cl, Cd-N(pyridyl), and Cd-N(oxime) stretching vibrations [54].

The specific spectral window of the FT-Raman spectra, where major spectral changes occur, is separately shown in Figure 7, right. These spectral changes are attributed to pyridine ring-breathing vibrations [49]. Compound phpaoH exhibits Raman peaks at 990 and 1032 cm^−1^ assigned to totally symmetric ring-breathing and trigonal ring deformation modes, respectively [50,55,56]. In both **1**∙H_2_O and **2**, a new Raman band appears at 1014 cm^−1^. Similar behavior has been observed (a) for complexes with the formulae NiCl_2_(py)_2_ and CoCl_2_(py)_2_ relative to the spectrum of pyridine (py), and (b) after adsorption of pyridine on transition metal electrodes [57]. A note is made of the fact that the Raman band at ~1000 cm^−1^ is characteristic of the mono-substituted benzene ring.

The ^1^H NMR spectra of **1**∙H_2_O and **2** in d_6_-DMSO are identical. The most remarkable feature is that the spectra are almost identical with the spectrum of free phpaoH in the same solvent. The spectra show a singlet signal at δ 11.61 ppm assigned to the hydroxyl proton [14,58,59] and a doublet at δ 8.49 ppm attributed to the proton of the aromatic carbon adjacent to the ring N-atom [14,24]. The other 2-pyridyl protons and the phenyl protons appear in the region δ 7.88–7.29 ppm. This experimental fact indicates that the two complexes decompose in solution, probably as indicated by equations (3) and (4). Strong evidence from our proposal comes from the molar conductivity values, Λ_M_ (10^−3^ M, 25 °C), for the two complexes in DMSO, which are 73 (**1**∙H_2_O) and 82 (**2**) S cm^2^ mol^−1^, indicative of 1:2 electrolytes [60]. The ^1^H-NMR spectra of **1**∙H_2_O and **2** in CD_3_OD are complicated, suggesting the presence of 2–3 species in equilibrium. For both complexes, at least one species contains coordinated phpaoH, as evidenced from the downfield shift of the doublet signal due to the proton of the aromatic carbon adjacent to the 2-pyridyl nitrogen atom, which appears at *δ* ~8.8 ppm.
(3)$$[{\mathrm{CdCl}}_{2}(\mathrm{phpaoH}{)}_{2}]\cdot H_{2}O+6d_{6}\text{-}\mathrm{DMSO}\overset{d6-\mathrm{DMSO}}{\to }[\mathrm{Cd}(d_{6}\text{-}\mathrm{DMSO}{)}_{6}]2^{+}+2{\mathrm{Cl}}^{-}+2\mathrm{phpaoH}+H_{2}O$$
(4)$$n\;\{[{\mathrm{CdCl}}_{2}(\mathrm{phpaoH})]{\}}_{n}+6nd_{6}\text{-}\mathrm{DMSO}\overset{d6-\mathrm{DMSO}}{\to }n[\mathrm{Cd}(d_{6}\text{-}\mathrm{DMSO}{)}_{6}]2^{+}+2n{\mathrm{Cl}}^{\text{-}}+n\;\mathrm{phpaoH}$$

## 3. Experimental Section

### 3.1. Materials and Physical-Spectroscopic Measurements

All manipulations were performed under aerobic conditions. Reagents and solvents were purchased from Alfa Aesar (Karlsruhe, Germany) and Aldrich (Tanfrichen, Germany) and used as received. The free ligand phenyl 2-pyridyl ketoxime (phpaoH) was synthesized as described in the literature [58] in a >80% yield; its purity was checked by ^1^H-NMR spectroscopy and the determination of its melting point (found, 148–149 °C; reported, 149–151 °C).

Elemental analyses (C, H, N) were performed by the University of Patras Center for Instrumental Analysis. Conductivity measurements were carried out at 25 °C with a Metrohm-Herisau E-527 bridge (Herisau, Switzerland) and a cell of standard constant. FT-IR spectra were recorded using a Perkin-Elmer 16PC spectrometer (Perkin-Elmer, Waltham, MA, USA) with samples in the form of KBr pellets. FT-Raman spectra were obtained using a Bruker (D) FRA-106/S component (Bruker, Karlsruhe, Germany) attached to an EQUINOX 55 spectrometer. An R510 diode-pumped Nd:YAG laser at 1064 nm was used for Raman excitation with a laser power 250 mW on the sample, utilizing an average of 100 scans at 4 cm^−1^ resolution. ^1^H NMR spectra were recorded on a 400 MHz Bruker Avance DPX spectrometer (Bruker AVANCE, Billerica, MA, USA) using (Me)_4_Si as an internal standard.

### 3.2. Syntheses of the Complexes

*[CdCl_2_(phpaoH)_2_]∙H_2_O (**1∙**H_2_O)*: A solution of phpaoH (0.079 g, 0.40 mmol) in Me_2_CO (6 mL) was added to an aqueous solution (6 mL) of CdCl_2_∙2H_2_O (0.043 g, 0.20 mmol). The resulting slurry was stirred for 20 min, filtered (to remove a small quantity of a white powder), and the filtrate was stored in a closed flask at room temperature. X-ray quality, colorless crystals of the product were obtained within 24 h. The crystals were collected by filtration, washed with EtOH (2 mL) and Et_2_O (2 × 2 mL), and dried in air. Yield: 31%. Anal. Calcd. (%) for C_24_H_22_N_4_CdCl_2_O_3_: C, 48.22; H, 3.72; N, 9.37. Found (%): C, 48.48; H, 3.58; N, 9.33. IR (KBr, cm^−1^): 3506m, 3384m, 3198wb, 3084w, 3060w, 3006w, 2850w, 1628m, 1590m, 1566w, 1470m, 1456s, 1442s, 1432sh, 1330m, 1294w, 1256m, 1186m, 1164m, 1122w, 1106w, 1072sh, 1054s, 1044sh, 1026s, 1014s, 958s, 922w, 912w, 898sh, 848m, 796s, 778m, 738w, 658m. Raman (cm^−1^): 3082sh, 3073sh, 3060s, 1642m, 1626m, 1594s, 1565s, 1470m, 1454m, 1328m, 1312w, 1292w, 1277w, 1257w, 1187m, 1160w, 1106w, 1053m, 1012m, 1002m, 799w, 775w, 735w, 636w, 612w, 582w, 405w, 327w, 307w, 233sh, 217w.^1^H-NMR (d_6_-DMSO, δ/ppm): 11.61(s,2H), 8.49(d,2H), 7.85(mt,4H), 7.39(mt,8H), 7.30(dd,4H), 3.35(s, protons of the lattice H_2_O and H_2_O contained in the deuterated solvent).

*{[CdCl_2_(phpaoH)]}_n_ (**2**):* A slurry of phpaoH (0.040 g, 0.20 mmol), Cd(ClO_4_)_2_∙6H_2_O (0.084 g, 0.20 mmol), and H_3_NOH^+^ Cl^−^ (0.028 g, 0.40 mmol) in Me_2_CO (5 mL) was stirred for 5 min. H_2_O (1 mL) was then added to the slurry, and the resulting pale yellow solution was stirred for a further 5 min and stored in a closed flask at room temperature. X-ray quality, colorless crystals of the product were precipitated after 10 d. The crystals were collected by filtration, washed with Et_2_O (3 × 1 mL), and dried in air. Yield: 79%. Anal. Calcd. (%) for C_12_H_10_N_2_OCdCl_2_: C, 37.77; H, 2.65; N, 7.34. Found (%): C, 37.95; H, 2.57; N, 7.50. IR (KBr, cm^−1^): 3388sb, 3056w, 3024w, 2896w, 1600m, 1540m, 1474m, 1448sh, 1436m, 1376m, 1328m, 1292w, 1260w, 1182m, 1162sh, 1102m, 1080sh, 1062sh, 1042s, 1022s, 954m, 928sh, 898w, 792m, 768w, 750w, 734w, 698s, 658m, 640w, 578w, 550w, 490w, 446w, 422w. Raman (cm^−1^): 3074sh, 3064s, 2925w, 1631m, 1597s, 1569s, 1474m, 1440w, 1430sh, 1378w, 1328m, 1293w, 1256w, 1184m, 1157w, 1101w, 1052w, 1015m, 1002m, 773w, 735w, 639w, 614w, 408w, 224w. ^1^H-NMR (d_6_-DMSO, δ/ppm): 11.60(sb,1H), 8.50(d,1H), 7.86(mt,2H), 7.39(mt,4H), 7.29(dd,2H).

### 3.3. Single-Crystal X-ray Crystallography

Colorless crystals of **1**∙H_2_O (0.12 × 0.16 × 0.49 mm) and **2** (0.04 × 0.09 × 0.27 mm) were taken from the mother liquor and immediately cooled to 160 K. Diffraction data were collected on a Rigaku R-AXIS Image Plate (Rigaku Americas Corporation, The Woodlands, TX, USA) diffractometer using graphite-monochromated Cu Kα radiation. Data collection (ω-scans) and processing (cell refinement, data reduction, and empirical absorption correction) were performed using the CrystalClear program [61]. The structures were solved by direct methods using SHELXS, ver. 2013/1 [62] and refined by full-matrix least-squares techniques on F^2^ with SHELXL, ver. 2014/6 [63]. All non-H atoms were refined anisotropically. The H atoms in the structure of **1**∙H_2_O were located by different maps and refined isotropically. The H atoms in the structure of **2** were introduced at calculated positions and refined as riding on their corresponding bonded atoms. Plots of the structures were drawn using the Diamond 3 program package [64]. Further crystallographic details for **1**∙H_2_O: 2θ_max_ = 130°; (Δ/σ)_max_ = 0.001; *R*_1_/*wR*_2_ (for all data) = 0.0276/0.0672. For **2**: 2θ_max_ = 130°; (Δ/σ)_max_ = 0.001; *R*_1_/*wR*_2_ (for all data) = 0.0325/0.0728.

Crystallographic data have been deposited with the Cambridge Crystallographic Data Center, Nos 1912320 (**1**∙H_2_O) and 1912319 (**2**). Copies of the data can be obtained free of charge upon application to CCDC, 12 Union Road, Cambridge, CB2 1EZ, UK: Telephone: +(44)-1223-762910; Fax: +(44)-1223-336033; E-mail: deposit@ccdc.cam.ac.uk, or via http:// www.ccdc.cam.ac.uk/conts/retrieving.html.

## 4. Concluding Comments and Perspectives

It is rather difficult to conclude on a project that has not been finished. We have partly fulfilled the goals mentioned in Section 1 (Introduction). With the drawbacks already mentioned (we have not worked with 2PC12 and 2PC14, which have been used in the extraction experiments but, instead, with phpaoH; we have not used H_2_O-CHCl_3_ solvent systems but, instead, H_2_O-Me_2_CO), we believe that complex 1∙H_2_O models satisfactorily the species [CdCl_2_(2PC12)_2_] and [CdCl_2_(2PC14)_2_] that have been proposed to form during the solvent extraction of the toxic Cd(II) from chloride-containing aqueous environments using the extractants 2PC12 and 2PC14. The characterization of 1 proves that neutral complexes [CdCl_2_(extractant)_2_] are capable of existence, favoring the transfer of the toxic metal ion into the organic phase. Our synthetic studies have also led to the 1:1 polymeric compound 2. The isolation of this complex, albeit with a different 2-pyridyl ketoxime than the real extractants, suggests that polymeric 1:1 CdCl_2_-2PC12 and CdCl_2_-2PC14 might exist. This explains the experimental fact that the extraction of Cd(II) increases upon an increase in the concentration of extractant [12]; it is obvious that such polymeric complexes should be avoided during the extraction process because their solubility in the organic phase might be low, disfavoring the % extraction.

From the synthetic inorganic chemistry viewpoint, our results emphasize the dramatic influence of the CdCl_2_:phpaoH molar ratio used on the product identity (monomeric vs. polymeric) and show that the structural chemistry of the CdCl_2_-phpaoH system is interesting in both molecular and supramolecular levels.

With the valuable knowledge obtained in this study, we have been trying to understand the molecular basis of other interesting phenomena that take place during the solvent extraction of toxic Cd(II) from Cl^−^-containing aqueous media. Among our future goals are: (1) The preparation and characterization of the species [Cd(NO_3_)_2_(extractant)_2_] and/or [CdCl(NO_3_)(extractant)_2_] that have been proposed to form during the Cd(II) extraction from aqueous solutions containing low chloride concentrations (<1.0 M) and high nitrate (>3.0 M) concentrations using 2PC12 and 2PC14; (2) the understanding of the negative effect of the increase of the chloride concentration in the aqueous phase on Cd(II) extraction by preparing anionic complexes, e.g., {(oximeH)[CdCl_3_]}_n_ and/or (oximeH)_2_[CdCl_4_]; (3) the study of the inability of 4-pyridyl ketoximes (e.g., 4PC12 and 4PC14, which are the analogs of 2PC12 and 2PC14, respectively, but with the oxime group on the position 4 of the pyridyl ring) by investigating CdCl_2_∙2H_2_O/4-pyridyl ketoxime complexes. Last, but not least, we have been using 2-pyridyl ketoximes with aliphatic substituents on the carbon atom for reactions with Cd(II); such ligands are more realistic models of the 2PC12 and 2PC14 extractants.

## Figures and Tables

**Figure 1 molecules-24-02219-f001:**
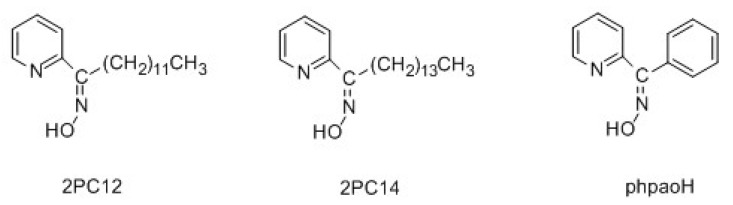
Structural formulae of the Cd(II) extractants, 2PC12, 2PC14, and the model ligand phenyl 2-pyridyl ketoxime (phpaoH) used in the present work.

**Figure 2 molecules-24-02219-f002:**
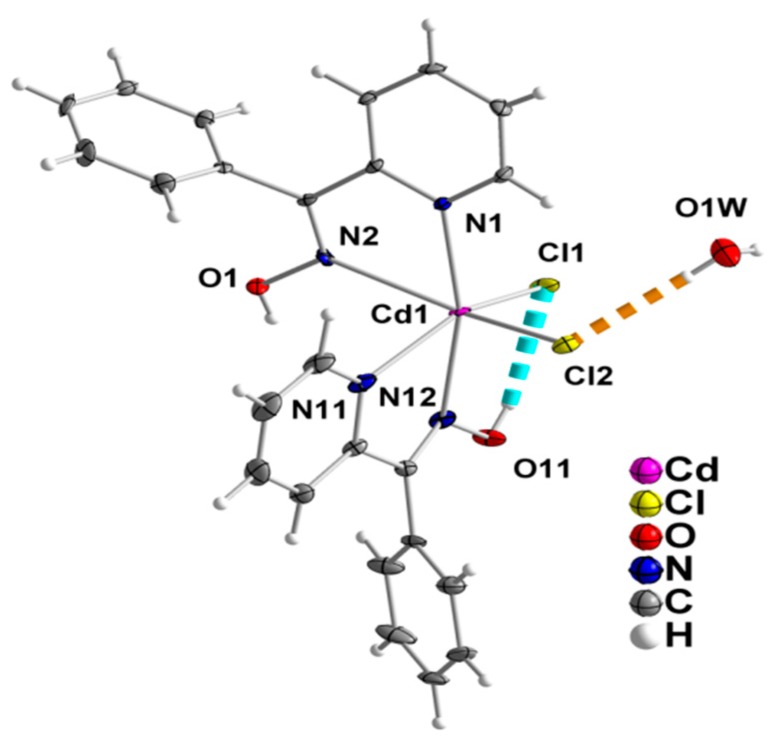
The molecules [CdCl_2_(phpaoH)_2_] and H_2_O that are present in the crystal structure of **1**∙H_2_O. The thermal ellipsoids are presented at the 50% level. Only diagnostic atoms have been numbered. Dashed lines indicate hydrogen bonds in which the symmetry operation is x, y, z. Through the hydrogen bond indicated with the orange color, molecular pairs of the type {[CdCl_2_(phpaoH)_2_] − H_2_O} are formed in the crystal.

**Figure 3 molecules-24-02219-f003:**
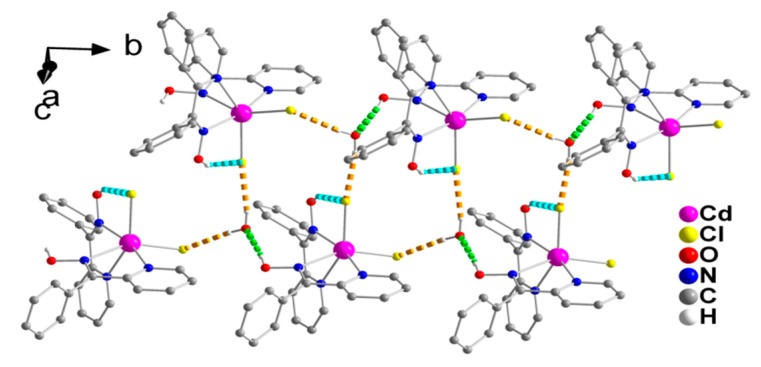
A small portion of one double chain, formed through hydrogen bonds, in the crystal structure of **1**∙H_2_O. The dashed light green and dashed cyan lines represent the hydrogen bonds O1-H(O1)∙∙∙O1W(x, −1 + y, z) and O11-H(O11)∙∙∙Cl1(x, y, z), respectively. The dashed orange lines represent the hydrogen bonds O1W-H_A_(O1W)-Cl2(x, y, z) and O1W-H_B_(O1W)∙∙∙Cl1(−x, 0.5 + y, 0.5 − z). For clarity, only hydrogen atoms involved in hydrogen bonding interactions are shown.

**Figure 4 molecules-24-02219-f004:**
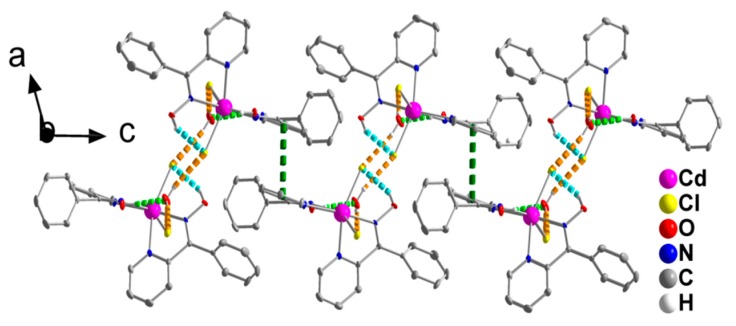
Formation of layers parallel to the (001) plane through π-π stacking interactions (dashed dark green lines) in the crystal structure of **1**∙H_2_O (see text for details). The view is presented along the b axis. The color code for the hydrogen bonds is the same as used in Figure 3.

**Figure 5 molecules-24-02219-f005:**
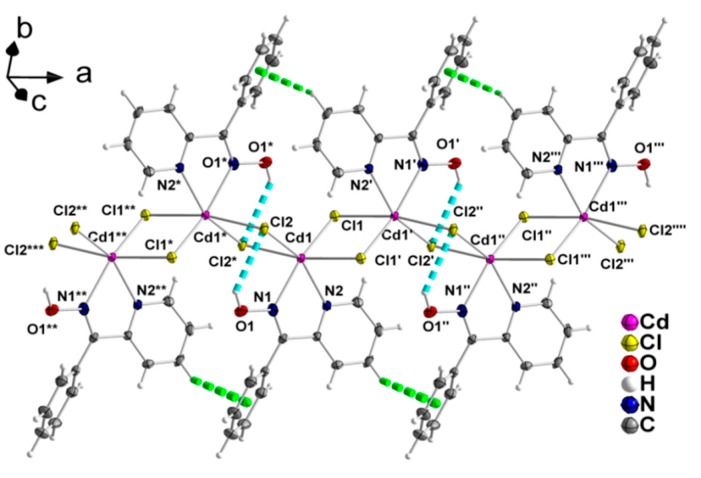
A small portion of one zigzag chain formed parallel to the ‘a’ axis in the crystal structure of **2**. The thermal ellipsoids are presented at the 50% level. The dashed cyan and light green lines represent O1-H(O1)∙∙∙Cl2 hydrogen bonds and C-H-π interactions [C3-Cg1 3.502(5) Å, C3-H(C3) 0.950(4) Å, H(C3)∙∙∙Cg1 2.952(1) Å, C3-H(C3)∙∙∙Cg1 118.4(3)°], respectively; C3 is a 2-pyridyl carbon atom and Cg1 is the centroid of the phenyl ring of phpaoH. Symmetry codes: (‘) –x + 2, −y + 2, −z + 1; (‘’) x + 1, y, z; (‘’’) –x + 3, −y + 2, −z + 1; (‘’’’) x + 2, y, z; (*) –x + 1, −y + 2, −z + 1; (**) –x, y, z; (***) –x, −y + 2, −z + 1.

**Figure 6 molecules-24-02219-f006:**
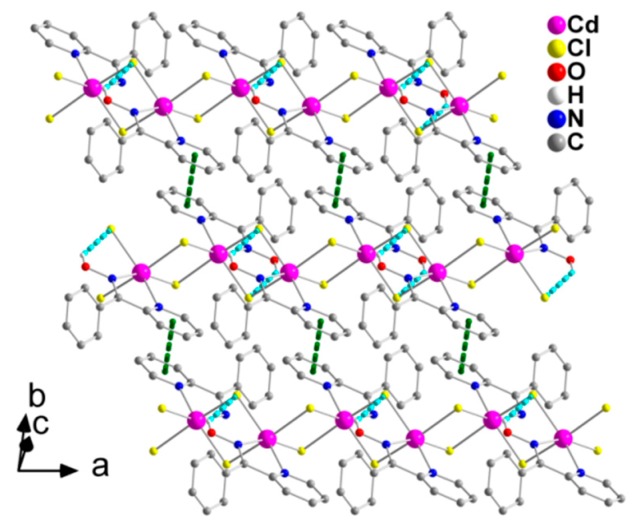
Formation of layers parallel to the (001) plane through π-π stacking interactions (dashed dark green lines) in the crystal structure of **2**. The C-H∙∙∙π interactions are not shown for clarity reasons. The color code for the hydrogen bonds is the same as used in Figure 5.

**Figure 7 molecules-24-02219-f007:**
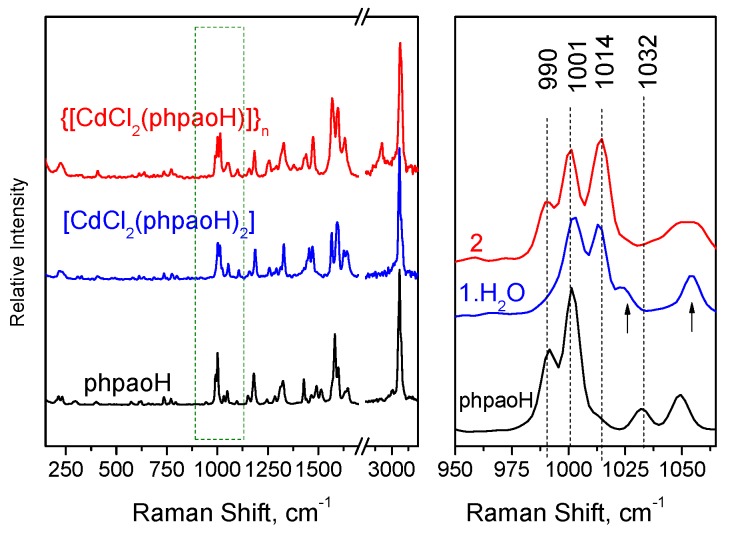
The FT-Raman spectra of compounds phpaoH, **1**∙H_2_O, and **2**.

**Table 1 molecules-24-02219-t001:** Crystallographic data and structural refinement parameters for complexes **1** and **2**.

Parameter	[CdCl_2_(phpaoH)_2_]∙H_2_O(1∙H_2_O)	{[CdCl_2_(phpaoH)]}_n_ (2)
Empirical formula	C_24_H_22_CdCl_2_N_4_O_3_	C_12_H_10_CdCl_2_N_2_O
Formula weight	597.75	381.52
Crystal system	monoclinic	triclinic
Space group	*P*2_1_/*c*	*P*-1
Color	colorless	colorless
Crystal size, mm	0.49 × 0.16 × 0.12	0.27 × 0.09 × 0.04
Crystal habit	block-shaped	block-shaped
*a,* Å	13.6684 (3)	7.0792 (1)
*b*, Å	8.8598 (2)	8.3260 (1)
*c*, Å	20.9270 (4)	12.3667 (3)
*α*, °	90.00	70.652 (1)
*β*, °	106.944 (1)	72.547 (1)
*γ*, °	90.00	85.323 (1)
Volume, Å^3^	2424.23 (9)	655.95 (2)
*Z*	4	2
Temperature, K	160	160
Radiation, Å	Cu-Kα (1.54178)	Cu-Kα (1.54178)
Calculated density, g∙cm^−3^	1.638	1.932
Absorption coefficient, mm^−1^	9.53	16.99
No. of measured, independent, and observed [*I* > 2*_σ_*(*I*)] reflections	36455, 4072, 3959	8385, 2014, 1855
*R* _int_	0.047	0.065
Number of parameters	394	164
Final *R* indices [*I* > 2*_σ_*(*I*)]^a^	*R*_1_ = 0.0267, *wR*_2_ = 0.0665	*R*_1_ = 0.0299, *wR*_2_ = 0.0715
Goodness-of-fit on *F*^2^	1.10	1.08
Largest differences peak and hole (e Å^−3^)	0.62/−1.26	0.68/−0.87

^a^*R*_1_ = Σ($\left|F_{0}\right|\;$−$\;\left|F_{c}\right|)$/Σ($\left|F_{0}\right|)$, *wR*_2_ = {Σ[*w*($F_{0}$^2^ −$\;F_{c}$^2^)^2^]/ Σ[*w*($F_{0}$^2^)^2^]}^1/2^, *w* = 1/[*σ*^2^($F_{0}$^2^) + (a*P)*^2^ + b*P*], where *P =* [max($F_{0}$^2^,0) + 2$F_{c}$^2^]/3 (a = 0.0307 and b = 3.3484 for **1**∙H_2_O; a = 0.0256 and b = 0.1966 for **2**).

**Table 2 molecules-24-02219-t002:** Selected bond lengths (Å) and angles (°) for complex[CdCl_2_(phpaoH)_2_]∙H_2_O (**1**∙H_2_O).

	**Bond Lengths (Å)**		**Bond Lengths (Å)**
Cd1-Cl1	2.532(1)	Cd1-N12	2.373(2)
Cd1-Cl2	2.534(1)	N2-O1	1.387(3)
Cd1-N1	2.323(2)	N12-O11	1.381(3)
Cd1-N2	2.509(2)	C6^a^-N2	1.286(3)
Cd1-N11	2.405(2)	C26^a^-N12	1.278(3)
	**Angles (°)**		**Angles (°)**
N1-Cd1-N2	69.8 (1)	N11-Cd1-N12	67.3 (1)
N1-Cd1-N11	105.3 (1)	N11-Cd1-Cl1	149.4 (1)
N1-Cd1-N12	166.2 (1)	N11-Cd1-Cl2	87.3 (1)
N1-Cd1-Cl1	102.1 (1)	N12-Cd1-Cl1	83.0 (1)
N1-Cd1-Cl2	92.3 (1)	N12-Cd1-Cl2	98.9 (1)
N2-Cd1-N11	80.2 (1)	Cl1-Cd1-Cl2	105.0 (1)
N2-Cd1-N12	98.9 (1)	C6^a^-N2-O1	112.9 (2)
N2-Cd1-Cl1	98.1 (1)	C26^a^-N12-O11	115.6 (2)
N2-Cd1-Cl2	152.3 (1)		

^a^ Atoms C6 and C26 (not labeled in Figure 2) are the oxime carbon atoms.

**Table 3 molecules-24-02219-t003:** Selected interatomic distances (Å) and angles (°) for complex {[CdCl_2_(phpaoH)]}_n_ (**2**)^a^.

	**Interatomic Distances (Å)**		**Interatomic Distances** **(Å)**
Cd1∙∙∙Cd1’	3.871 (1)	Cd1-Cl2*	2.686 (1)
Cd1∙∙∙Cd1*	3.978 (1)	Cd1-N1	2.407 (4)
Cd1-Cl1	2.564 (1)	Cd1-N2	2.323 (3)
Cd1-Cl1’	2.682 (1)	C6^b^-N1	1.293 (5)
Cd1-Cl2	2.575 (1)	N1-O1	1.386 (4)
	**Angles (°)**		**Angles (°)**
N1-Cd1-N2	67.6(1)	Cl1-Cd1-Cl2	112.6(1)
N1-Cd1-Cl1	160.1(1)	Cl1-Cd1-Cl2*	87.5(1)
N1-Cd1-Cl1’	101.4(1)	Cl1’-Cd1-Cl2	93.1(1)
N1-Cd1-Cl2	86.1(1)	Cl1’-Cd1-Cl2*	168.4(1)
N1-Cd1-Cl2*	88.7(1)	Cl2-Cd1-Cl2*	81.8(1)
N2-Cd1-Cl1	97.8(1)	Cd1-Cl1-Cd1’	95.1(1)
N2-Cd1-Cl1’	83.3(1)	Cd1-Cl2-Cd1*	98.3(1)
N2-Cd1-Cl2	152.0(1)	C6^b^-N1-O1	114.5(4)
N2-Cd1-Cl2*	106.2(1)	Cd1-N1-O1	123.9(2)
Cl1-Cd1-Cl1’	84.9(1)	Cd1-N1- C6^b^	120.5(3)

^a^ Symmetry codes: (‘) −x + 2, −y + 2, −z + 1; (*) −x + 1, −y + 2, −z + 1. ^b^ Atom C6 (not labeled in Figure 5) is the oxime carbon atom.

**Table 4 molecules-24-02219-t004:** Hydrogen bonding interactions (Å, deg) in the crystal structure of [CdCl_2_(phpaoH)_2_]∙H_2_O (**1**∙H_2_O).

D-H∙∙∙A	d(D∙∙∙A)	d(D-H)	d(H∙∙∙A)	<DHA	Symmetry Code of A
O11-H(O11)∙∙∙Cl1	3.098(2)	0.77(4)	2.38(4)	155(3)	x, y, z
O1W-H_A_(O1W)∙∙∙Cl2	3.122(3)	0.80(5)	2.33(5)	174(4)	x, y, z
O1W-H_B_(O1W)∙∙∙Cl1	3.241(2)	0.85(4)	2.41(5)	165(4)	−x, 0.5+y, 0.5−z
O1-H(O1)∙∙∙O1W	2.623(3)	0.76(3)	1.87(3)	170(3)	x, −1+y, z

D = donor; A = acceptor.

**Table 5 molecules-24-02219-t005:** Hydrogen bonding interactions (Å, deg) in the crystal structure of {[CdCl_2_(phpaoH)]}_n_ (**2**).

D-H∙∙∙A	d(D∙∙∙A)	d(D-H)	d(H∙∙∙A)	<DHA	Symmetry Code of A
O1-H(O1)∙∙∙Cl2	3.379(3)	0.84	2.69	140	x, y, z
O1-H(O1)∙∙∙Cl1*	3.378(3)	0.84	2.74	133	−x + 1, −y + 2, −z + 1

D = donor; A = acceptor.

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
