# Peer review of "Modeling the Solvent Extraction of Cadmium(II) from Aqueous Chloride Solutions by 2-pyridyl Ketoximes: A Coordination Chemistry Approach"

_molecules, 2019, doi:10.3390/molecules24122219_

Round 1
Reviewer 1 Report
The presented manuscript describes obtaining and characterization of two Cd compounds as model structures of complexes which may be formed during Cd removal by extraction technique. The manuscript is clearly written and contains new information about crystal structure and bonding of anhydrous and hydrated Cd compounds. The authors in the conclusion state that work has not been finished yet. And they studied only model compound instead of compounds with practically used extractants.
It would be nice if authors may provide more evidences that phpaoH has similarities with 2PC12 and 2PC14. Because -(CH2)11CH3 and -(CH2)12CH3 are slightly different with phenyl groups. Is there any analogy with complexes with other d-metals, for example?
Hydrated and anhydrous compound were obtained using different Cd sources and different media. Is it possible to estimate the role of ClO4- ion and hydroxylamine hydrochloride for the resulting (CdCl2(phpaoH))n formation? From the other side as follows from the text the (CdCl2(phpaoH))n was obtained from solution with small water containing.
Author Response
<< The presented manuscript describes obtaining and characterization of two Cd compounds as model structures of complexes which may be formed during Cd removal by extraction technique. The manuscript is clearly written and contains new information about crystal structure and bonding of anhydrous and hydrated Cd compounds. The authors in the conclusion state that work has not been finished yet. And they studied only model compound instead of compounds with practically used extractants. >>
The reviewer gives a brief description of our work.
<< It would be nice if authors may provide more evidences that phpaoH has similarities with 2PC12 and 2PC14. Because -(CH2)11CH3 and -(CH2)12CH3 are slightly different with phenyl groups. Is there any analogy with complexes with other d-metals, for example? >>
The comment is correct. We had described the drawback of using phpaoH for our modelling studies in sections 1 (Introduction) and 4 (Concluding Comments and Perspectives) of the originally submitted ms. To satisfy Reviewer's 1 concern, we discuss the analogy with complexes of other metal ions by adding a new sentence ("Our previous experience on ..... our model studies") in the end of section 1 in the revised version of the ms. Since there are no previously reported structural studies with the -(CH2)11CH3- and -(CH2)13CH3-containing 2-pyridyl ketoximes, we compare previously reported complexes containing methyl 2-pyridyl ketoxime (mepaoH) and phenyl 2-pyridyl ketoxime (phpaoH) which contain a short aliphatic substituent (-CH3) and an aromatic substituent (-C6H5; phpaoH) on the oxime carbon atom respectively; the comparison shows that the two ligands behave similarly in many cases despite the different electronic and steric characteristics. Thus, phpaoH can be considered as a reliable analogue of the real extractants (2PC12, 2PC14).
<< Hydrated and anhydrous compound were obtained using different Cd sources and different media. Is it possible to estimate the role of ClO4- ion and hydroxylamine hydrochloride for the resulting (CdCl2(phpaoH))n formation? >>
The comment is absolutely logical. To answer the question of the reviewer, we have added a new sentence ("With the results at hand, ...... crystals obtained") in part 2.1 (Synthetic Comments) of the revised ms, where we briefly discuss the possible role of ClO4- and hydroxylamine for the formation of the coordination polymer.
<< From the other side as follows from the text the (CdCl2(phpaoH))n was obtained from solution with small water containing. >>
Again the comment is scientifically correct. To address this point, we have added a new sentence ("Use of a solvent mixture ...... crystals worse") in part 2.1 (Synthetic Comments) of the revised ms, where we briefly discuss the use of a solvent mixture with a lower (than in the other preparations) H2O volume percentage.
WE THANK REVIEWER 1 FOR HER/HIS VALUABLE COMMENTS, THE ANSWERS OF WHICH IN THE REVISED MS HELP US TO IMPROVE THE QUALITY OF THE WORK.
Reviewer 2 Report
Reviewer Comment_Manuscript Number: Molecules-515549
Title: Modelling the Solvent Extraction of Cadmium(II) from Aqueous Chloride Solutions by 2-pyridyl Ke-toximes: An Inorganic Chemistry Approach
Type:
Article
The authors have reported an on-going study/project, which is a good idea, however, I have some information to draw their attention as shown below.
Abstract:
I think this section is what readers would be interesting in order to get the basic knowledge of the study, however, the writing is too lengthy, the authors should revise this section.
Keywords:
This is also more than the required number, author should revise and be specific for proper indexing.
Introduction:
Solvents abbreviation should be explained properly e.g. organic solvents (EtOH, MeCN, CHCl3) or organic solvent mix-tures (EtOH/MeCN, MeCN/CHCl3),
What could be the problem associated with base addition, as recorded ....”we thus avoided the addition of an external base (e.g., LiOH, Et3N, R4NOH, etc.) in the reaction systems”.
Results and Discussion:
Under the Spectroscopic Studies, within the FT-IR below reference should be cited:
Ejidike, I.P. Cu(II) Complexes of 4-[(1E)-N-{2-[(Z)-benzylidene-amino]ethyl}ethanimidoyl]benzene-1,3-diol Schiff base: Synthesis, spectroscopic, in-vitro antioxidant, antifungal and antibacterial studies. Molecules 2018, 23, 1581; doi:10.3390/molecules23071581.
New bonds identified like Cd-O, Cd-N and Cd-Cl where not given any information in the explanation although they identified at the Syntheses of the complexes section.
Experimental Section
Since the study is based on modelling the solvent extraction of cadmium(II) from aqueous chloride solutions by 2-pyridyl ke-toximes, there should be a section to investigate the extraction procedures. How the extraction was carried out to justify the desired title and to show that the extraction was the principle focus, not complexation reaction. The study is more like ligand-metal interaction leading to crystal formation.
The above statement can be linked to the authors Concluding Comments and Perspectives section about “This explains the experimental fact that the extraction of Cd(II) increases upon increase of the concentration of extractant [12]; it is obvious that such polymeric complexes should be avoided during the extraction process because their solubility in the organic phase might be low, disfavoring the % extraction”
With this, % extraction study will be sufficient to confirm the topic as proposed by the authors.
Concluding Comments and Perspectives section
The statement “With the valuable knowledge obtained in this study, we have been trying to understand the molecular basis of other interesting phenomena mentioned in ref. [12].” Should re-phrased to eliminate the inclusion of reference in the conclusion.
References:
References should be cross-checked properly for correctness in line-with the journal style.
Author Response
<< The authors have reported an on-going study/project, which is a good idea, however, I have some information to draw their attention as shown below. >>
We thank the reviewer for her/his overall positive view of our work.
Abstract:
<< I think this section is what readers would be interesting in order to get the basic knowledge of the study, however, the writing is too lengthy, the authors should revise this section. >>
We agree with the reviewer's comment. We have condensed the "Abstract".
Keywords:
<< This is also more than the required number, author should revise and be specific for proper indexing. >>
The comment is correct. We have deleted three(3) keywords in the revised version of the ms.
Introduction:
<< Solvents abbreviation should be explained properly e.g. organic solvents (EtOH, MeCN, CHCl3) or organic solvent mix-tures (EtOH/MeCN, MeCN/CHCl3), >>
The explanation (i.e. full name) of the solvents abbreviation has been given in the text of the revised ms at the first point of their appearance.
<< What could be the problem associated with base addition, as recorded ....”we thus avoided the addition of an external base (e.g., LiOH, Et3N, R4NOH, etc.) in the reaction systems”. >>
We had mentioned in detail this point in part 2.1 (Synthetic Comments) of the originally submitted ms. Since the extractants 2PC12 and 2PC14 are in their neutral form (pH = 3.5-3.8) in the real extraction processes reported in ref. [12], we targeted the preparation of model Cd(II) complexes in which phpaoH would be in its neutral form. Since the addition of external bases could easily lead to complexes with the deprotonated ligand (phpao-), we avoided the participation of external bases in the reaction mixtures.
Results and Discussion:
<< Under the Spectroscopic Studies, within the FT-IR below reference should be cited: Ejidike, I.P. Cu(II) Complexes of 4-[(1E)-N-{2-[(Z)-benzylidene-amino]ethyl}ethanimidoyl]benzene-1,3-diol Schiff base: Synthesis, spectroscopic, in-vitro antioxidant, antifungal and antibacterial studies. Molecules 2018, 23, 1581; doi:10.3390/molecules23071581. >>
Following the suggestion by the reviewer, we have added the reference in the revised version of the ms. The new reference is ref. [49]; the numbering scheme of the references after this addition has been modified accordingly.
<< New bonds identified like Cd-O, Cd-N and Cd-Cl where not given any information in the explanation although they identified at the Syntheses of the complexes section. >>
We assume that the referee means that we have not discussed the spectroscopic features of the Cd-N and Cd-Cl bonds that are present in the two complexes. The remark is scientifically correct. We have added a new sentence ("The Raman peaks ........ stretching vibrations [54]") in part 2.3 (Spectroscopic Studies) of the revised version of the ms where we discuss the stretching vibrations of the Cd-Cl and Cd-N bonds; exact assignments are not possible. We have added the new ref. [54] to support the text; the numbering scheme of the references after this addition has been modified accordingly.
Experimental Section:
<< Since the study is based on modelling the solvent extraction of cadmium(II) from aqueous chloride solutions by 2-pyridyl ke-toximes, there should be a section to investigate the extraction procedures. How the extraction was carried out to justify the desired title and to show that the extraction was the principle focus, not complexation reaction. The study is more like ligand-metal interaction leading to crystal formation.
The above statement can be linked to the authors Concluding Comments and Perspectives section about “This explains the experimental fact that the extraction of Cd(II) increases upon increase of the concentration of extractant [12]; it is obvious that such polymeric complexes should be avoided during the extraction process because their solubility in the organic phase might be low, disfavoring the % extraction”
With this, % extraction study will be sufficient to confirm the topic as proposed by the authors. >>
We were clear and completely honest when we submitted our work. The work is a modelling study based on the extraction procedures following ref. [12]. Thus, there is no need for a new section to investigate the extraction procedures. We have never claimed that we performed extraction experiments! The title refers to "Modelling the Solvent Extraction of Cadmium(II) ........". In addition, our Laboratory (which is a synthetic-and not analytical-laboratory) is not equipped for extraction studies and we do not have the expertise for such studies! We have provided the reader with the brief extraction results (based on ref. [12]) in section 1 (Introduction), so she/he can easily understand the flow of the text. To show that we respect the opinion of the reviewer, we have slightly changed the title of the ms from ".... . An Inorganic Chemistry Approach." to ".... . A Coordination Chemistry Approach." The latter is more realistic and closer to our synthetic/characterization studies. This change has been also made in the text.
Concluding Comments and Perspective section:
<< The statement “With the valuable knowledge obtained in this study, we have been trying to understand the molecular basis of other interesting phenomena mentioned in ref. [12].” Should re-phrased to eliminate the inclusion of reference in the conclusion. >>
The comment is correct. We have rephrased the relevant sentence eliminating the reference.
References:
<< References should be cross-checked properly for correctness in line-with the journal style. >>
We have cross-checked properly the correctness of the references and these are in line with the journal's style.
WE THANK REVIEWER 2 FOR HER/HIS CRITICAL COMMENTS AND SUGGESTIONS, WHICH HAVE BEEN INCORPORATED IN THE REVISED MS HELPING THE IMPROVEMENT OF ITS QUALITY.
Reviewer 3 Report
The article titled "Modelling solvent ... Approach" by Mazarakioti et al. describes the synthesis, structural and spectroscopic study of two coordination complexes of Cd(II) with 2-pyridyl ketoxime ligands. The authors have used phenyl 2-pyridyl ketoxime as an approximate model to extractants containing long alkaline chains. The authors have also described how the molar ratio is a crucial factor in determining which complex is obtained. Crystal structures, spectroscopic studies presented are sound and support their conclusions. I would recommend acceptance of this article in Molecules requiring some minor corrections to the manuscript as follows:
1. It is unnecessary to write “µ2-chloro” as “µ” itself means that the ligand binds two metal centers. A number is needed only when more than two metal centers are bound to a ligand, e.g., µ3-oxo, µ4-Cl, etc.
2. What do the authors mean by “metal values” in page 2, line 2, 2nd paragraph?
3. Page 8, 2nd line below the tables: change “a crystallographic axis” to “crystallographic a axis”.
4. I would also italicize all axes in the article.
5. The caption of Fig. 5 states “dashed cyan lines…”, but no such line is shown in the figure.
6. For 2, the centroid-centroid distance of >4 Å seems a little longer than what is expected of a typical π-π interaction; is there any C-H…π interaction present in the structure?
7. Page 10, 7th line in 2nd paragraph: please change the sentence “there have been reported only two mononuclear complexes…” to “there have been only two mononuclear complexes reported…”
Author Response
<< The article titled "Modelling solvent ... Approach" by Mazarakioti et al. describes the synthesis, structural and spectroscopic study of two coordination complexes of Cd(II) with 2-pyridyl ketoxime ligands. The authors have used phenyl 2-pyridyl ketoxime as an approximate model to extractants containing long alkaline chains. The authors have also described how the molar ratio is a crucial factor in determining which complex is obtained. Crystal structures, spectroscopic studies presented are sound and support their conclusions. I would recommend acceptance of this article in Molecules requiring some minor corrections to the manuscript as follows: >>
We thank very much the reviewer for her/his positive and warm comments.
<< 1. It is unnecessary to write “µ2-chloro” as “µ” itself means that the ligand binds two metal centers. A number is needed only when more than two metal centers are bound to a ligand, e.g., µ3-oxo, µ4-Cl, etc. >>
The comment is correct. We have been using "μ" instead of "μ2" throughout the revised version of the ms.
<< 2. What do the authors mean by “metal values” in page 2, line 2, 2nd paragraph? >>
Indeed the term "metal values" was not clear. We have changed it to "metal contents".
<< 3. Page 8, 2nd line below the tables: change “a crystallographic axis” to “crystallographic a axis”. >>
Correct. The change has been performed.
<< 4. I would also italicize all axes in the article. >>
We have followed the correct suggestion by the reviewer.
<< 5. The caption of Fig. 5 states “dashed cyan lines…”, but no such line is shown in the figure. >>
The comment is absolutely correct. We have replaced the old Figure 5 with a new one in which the dashed cyan lines are shown. In addition, the new Figure 5 also shows the C-H∙∙∙π interactions (light green lines), please see the below comment (comment 6) of the reviewer. The caption of the new Figure 5 has been modified accordingly.
<< 6. For 2, the centroid-centroid distance of >4 Å seems a little longer than what is expected of a typical π-π interaction; is there any C-H…π interaction present in the structure? >>
The comment is scientifically correct. Indeed, looking carefully at the structure we discovered that there are C-H∙∙∙π interactions with a pyridyl carbon atom as donor and the phenyl ring of phpaoH as acceptor. We have added a sentence ("In addition, there is a C-H∙∙∙π interaction........acceptor) in the text of the revised ms to indicate the existence of such interactions. These interactions are shown with light green lines in the new Figure 5, and its revised caption mentions these interactions. In Figure 6, which shows the formation of the layers in the structure of compound 2 we prefer not to draw the C-H∙∙∙π interactions for clarity reasons and we indicate this by adding a sentence ("The C-H∙∙∙π interactions........clarity reasons) in its revised caption.
<< 7. Page 10, 7th line in 2nd paragraph: please change the sentence “there have been reported only two mononuclear complexes…” to “there have been only two mononuclear complexes reported…” >>
We have changed the sentence as correctly requested by the reviewer.
WE THANK REVIEWER 3 FOR HER/HIS VALUABLE COMMENTS AND CORRECTIONS, WHICH HELP US TO IMPROVE A LOT THE QUALITY AND THE READABILITY OF THE WORK.
Round 2
Reviewer 2 Report
The authors have done a wonderful job by improving the quality of the present manuscript.
I give credits to this team of researcher for their efforts and cooperativeness.